# Discrepancies Between the Tennessee Nomogram and Oncotype DX: Implications for the Korean Breast Cancer Population—The BRAIN Study

**DOI:** 10.3390/cancers17183083

**Published:** 2025-09-21

**Authors:** Suk Jun Lee, Joo Heung Kim, Jee Hyun Ahn, So Hyeon Gwon, Ilkyun Lee, Seho Park, Nak-Hoon Son

**Affiliations:** 1Department of Surgery, Catholic Kwandong University College of Medicine, International St. Mary’s Hospital, Incheon 22711, Republic of Korea; atsukjun@hanmail.net (S.J.L.); iklee@ish.ac.kr (I.L.); 2Department of Surgery, Yongin Severance Hospital, Yonsei University College of Medicine, Yongin 16995, Republic of Korea; pianac@yuhs.ac; 3Division of Breast Surgery, Department of Surgery, Yonsei University College of Medicine, Seoul 03722, Republic of Korea; jhahn35@yuhs.ac; 4Department of Statistics, Keimyung University, Daegu 42601, Republic of Korea; gwonsohyeon@kmu.kr

**Keywords:** breast cancer, Korean population, Oncotype Dx, Tennessee nomogram, validation, discordance

## Abstract

Breast cancer treatment decisions often rely on the Oncotype DX (ODX) test, which predicts the risk of cancer recurrence and the potential benefit of chemotherapy. However, this genetic test is expensive and not available to all patients worldwide. The Tennessee nomogram is a simpler, clinicopathology-based tool designed to approximate ODX results. We examined whether this nomogram could accurately predict ODX risk scores in Korean patients with early-stage, hormone receptor-positive, HER2-negative breast cancer. In our study of 1298 patients, the nomogram accurately predicted low-risk results for most individuals but underestimated risk in some patients with aggressive tumor features, such as high tumor grade, lack of progesterone receptor expression, and high Ki-67 levels. These patients were often assigned a low-risk score by the nomogram despite having high ODX scores, particularly in the borderline-high range (25–30). Our findings suggest that while the Tennessee nomogram may help guide treatment decisions when genetic testing is unavailable, clinicians should exercise caution when tumors exhibit aggressive characteristics and consider performing ODX testing to ensure optimal therapy selection.

## 1. Introduction

Oncotype Dx (ODX) is widely used for patients with hormone receptor (HR)-positive and human epidermal growth factor 2 (HER2)-negative breast cancer. ODX provides prognostic and predictive information by assessing the risk of recurrence [1,2]. The Trial Assigning Individualized Options for Treatment (TAILORx), initiated in 2006, categorizes patients into three groups based on their Recurrence Score (RS) [3,4]. Patients with RS < 11 are classified as low risk, those with RS between 11 and 25 are classified as intermediate risk, and individuals with RS > 25 are classified as high risk [5,6]. The ODX test indicates that patients in the high-risk group benefit significantly from chemotherapy [7], whereas those in the low-risk group do not experience additional benefits compared to those in endocrine therapy alone [8]. However, the effectiveness of chemotherapy for intermediate-risk patients remains uncertain [6,9,10,11].

Despite its proven clinical utility, the Oncotype DX (ODX) test remains costly (approximately $4000), and although utilization rates in the United States have risen, they remain incomplete. Recent national data indicate that between 43% and 48% of eligible early-stage HR+/HER2− breast cancer patients underwent ODX testing in the U.S. between 2010 and 2017 [12,13,14,15]. In Europe, comprehensive population-level data are scarce; however, observational data from Italy suggest its real-world use is significant yet variably implemented across regions [16]. These findings highlight a gap between the clinical value of ODX and its accessibility in actual practice, underscoring the need for alternative, more accessible tools to guide treatment decisions [17].

A study from the University of Tennessee Medical Center developed a nomogram using six clinicopathological variables derived from large-scale information from the National Cancer Database to predict ODX results in ER-positive, HER2-negative, and patients with lymph node-negative breast cancer. Tested on 27,685 original and 12,763 external validation cohorts, the nomogram demonstrated high accuracy [18,19]. However, this model is limited because it was developed based on data from American patients [20].

Therefore, this study aims to validate whether the nomogram developed by the University of Tennessee Medical Center is applicable to the Korean breast cancer patient dataset and explore its limitations as well as potential means of improvement.

## 2. Materials and Methods

### 2.1. Study Population

We retrospectively reviewed data from patients with HR(+)/HER2(−) node-negative breast cancer who underwent breast cancer surgery and had ODX testing at Severance Hospital, Yonsei University College of Medicine, Seoul, South Korea, between May 2013 and August 2023. Patients were excluded if they did not undergo multigene testing, had tumors > 5 cm, had node-positive disease, or had other histologic types except invasive ductal or lobular carcinoma. Overall, 1298 patients were included in the analysis. This study was approved by the Institutional Review Board of Yonsei University College of Medicine, Seoul, South Korea (Approval No. 4–2023−1639, approval date: 8 February, 2024), but the requirement for informed consent was waived owing to the retrospective study design.

### 2.2. Clinicopathologic Evaluation

We collected basic patient information, such as age and clinicopathological characteristics, including tumor size, histologic type, histologic grade, progesterone receptor status, HER2 status, Ki-67 index value, and ODX recurrence risk. The tumor stage was assessed according to the 8th edition of the American Joint Committee on Cancer (AJCC) TNM staging system [21].

The cutoff values for low- and high-risk RS were determined based on the results of the TAILORx clinical trial, with 0–25 and 26–100 considered low and high risk, respectively [10]. The trial indicated that patients aged < 50 years with RS between 16 and 25 could benefit from adjuvant chemotherapy. In contrast, we reclassified the RS values, designating 0–25 as the low-risk group and 26–100 as the high-risk group [4].

### 2.3. Model Validation and Statistical Analysis

For the analysis of categorical data, the chi-square test and Fisher’s exact test were used based on the expected frequency. The results are expressed as frequencies and percentages. Logistic regression analysis was used to obtain the probability (score) of the risk groups of Korean patients with breast cancer based on the Tennessee Nomogram. The same criteria as those used in the Tennessee Nomogram were applied to calculate the predictive probability (score). The predicted probability for each patient was then dichotomized into high- versus low-risk groups using a 0.5 threshold, consistent with the original Tennessee Nomogram’s classification approach.

To identify the confusion matrix for the predicted risk groups of the Tennessee Nomogram and the ODX RS and to evaluate the predicted performance of the Tennessee Nomogram, we presented the sensitivity, specificity, accuracy, positive predictive value (PPV), negative predictive value (NPV), and area under the receiver operating characteristic (ROC) curve with area under the curve (AUC) and 95% confidence interval (CI). Additionally, a histogram of the ODX RS was presented to determine the distribution of the predicted risk groups according to the ODX RS. Various metrics to evaluate model performance can be differently applied depending on the clinical context [22].

Statistical analyses were performed using SAS (version 9.4; SAS Institute) and R (version 4.4.1; R Foundation for Statistical Computing, Vienna, Austria).

## 3. Results

### 3.1. Patient Characteristics

Table 1 presents the descriptive clinicopathological characteristics of the patients. The numbers of patients in the low- and high-risk groups were 1105 (85.1%) and 193 (14.9%), respectively. The proportion of patients under the age of 50 years in our study cohort (50.6%) was higher than that in the Tennessee model cohort (25.6%). However, the average tumor sizes in the two studies were 16.0 and 16.1 mm, respectively, showing no substantial difference. Aside from the age distribution, the clinicopathological characteristics in this study were similar to those of the Tennessee model study population [18].

### 3.2. Accuracy of the Tennessee Nomogram in the Korean Population

To evaluate the accuracy of the Tennessee nomogram, we compared the predictive results of the nomogram with the actual ODX results (Figure 1). The performance metrics of the model were as follows: sensitivity, 0.130 (95% CI: 0.082–0.177); specificity, 0.989 (95% CI: 0.983–0.995); accuracy, 0.861 (95% CI: 0.843–0.88); PPV, 0.676 (95% CI: 0.525–0.827); NPV, 0.867 (95% CI: 0.848–0.886); and AUC, 0.776 (95% CI: 0.740–0.814). The overall accuracy of the nomogram was 86.1%, which was 89% in the original study of the Tennessee nomogram [18] but 85% was reported in another validation study in Korea [23].

### 3.3. Characteristics of Discordant Group

In the subgroup analysis, patient characteristics were grouped into four: TP, FN, FP, and TN. Additional analysis was conducted to examine the FN and FP groups, where the actual Oncotype results differed from the nomogram predictions. To investigate the potential reasons for prediction failure, we compared these two groups with the TP and TN groups. The distribution of clinicopathological features was specifically analyzed by comparing FN with TP and FP with TP groups.

Table 2 shows the comparisons of the clinicopathological characteristics between the FN and TP groups. The FN group had lower histological grades than the TP group (*p* < 0.001). A higher progesterone receptor (PR) positivity rate was observed in the FN group than in the TP group (*p* < 0.001). Ki-67 levels below 20% were more predominant in the FN group than in the TP group (*p* = 0.017).

Table 3 shows the comparisons of the clinicopathological characteristics between the FN and TN groups. The mean age was higher in the FN group than in the TN group (*p* < 0.001). The FN group had a higher histological grade than the TN group (*p* < 0.001). The FN group exhibited a lower PR-positivity rate than the TN group (*p* < 0.001). Ki-67 levels were higher in the FN group than in the TN group (*p* < 0.001).

Figure 2 shows the ODX risk score distributions in the FN and FP groups. Most patients (52.38%) in the FN group were clustered within the ODX risk score range of 25–30, whereas the majority of patients (58.33%) in the FP group had ODX risk scores between 21–and 25.

## 4. Discussion

The Tennessee nomogram correctly classified 25 patients (1.9%) as high risk (true positive, TP) and 1093 patients (84.2%) as low risk (true negative, TN). However, 180 patients (13.9%) exhibited discordant findings between the nomogram and actual ODx results. Additionally, 12 patients (0.9%) were predicted as high risk using the Tennessee model but belonged to the actual ODx low-risk group (false positive, FP). Furthermore, 168 patients (12.9%) were incorrectly identified as low risk despite belonging to the actual ODx high-risk group (false negative, FN).

This study validated the applicability of the Tennessee nomogram to clinical data from Korean patients, showing a predictive accuracy of 86.1%. Although this is similar to the C-indices of 89% and 85% reported in the Tennessee nomogram [18] and previous Korean validation study [23], respectively, several factors contribute to these discrepancies. One key difference is the age distribution. In this study, the mean patient age was 51.0 ± 9.9 years, whereas in the Tennessee study, it was 59.3 ± 10.3 years. The proportion of invasive ductal carcinoma (IDC) in this study was higher than that in the Tennessee study. This age-distribution shift can alter the pretest probability of high RS and the effect sizes of key predictors (e.g., grade, PR, Ki-67), potentially affecting both calibration and discrimination. Younger patients are more likely to present with higher proliferative indices and lower PR expression, which could reduce the transportability of the model to our population [6,24,25,26]. Age-stratified calibration/discrimination analyses, reconsideration of thresholds, and—if needed—recalibration or model updating (including incorporation of Ki-67) would improve predictive accuracy and clinical utility [27].

Furthermore, population-specific factors in Asian cohorts may also contribute to the observed differences. In particular, East Asian breast cancers have been reported to exhibit distinct mutational patterns, including a higher prevalence of PIK3CA mutations and a lower frequency of GATA3 mutations [28]. These molecular and systemic disparities highlight the importance of validating and, if necessary, recalibrating predictive tools in different populations. Moreover, variation in the expression of proliferation-related genes across ethnic groups may further explain the reduced accuracy observed in our study.

FN and FP groups are essential for evaluating model performance. However, we specifically focused on the FN group because it is more likely to be exposed to greater risk by losing the chance to receive adjuvant chemotherapy. Subgroup analysis revealed that the FN group was considerably different in age, histological grade, PR positivity rate, and Ki-67 levels (Table 2 and Table 3). Besides the variables included in the Tennessee model, the Ki-67 level was the only variable that showed a statistically significant difference in the FN group. Further investigation is required to study the influence of Ki-67 levels on model performance.

The ODX results in the discordant groups were mostly distributed between 21 and 30. In the FN group, the ODX results ranged between 25 and 30. This is because no intermediate-risk group for ODX was categorized at the time the Tennessee model was developed [5,18]. This potentially necessitates the development of a new model that categorizes cases into low-, intermediate-, and high-risk groups [29]. Notably, our data indicate that the Tennessee model predicts outcomes with reasonable accuracy when the actual high-risk ODX score is ≥30. However, its accuracy is markedly reduced between 25 and 30, particularly in patients with a higher histological grade, PR negative status, and higher Ki-67 levels in the predicted low-risk group. In such circumstances, performing an ODX test may be more beneficial than relying on a nomogram for prediction.

Given that the Tennessee nomogram is presented as a lower-cost alternative to the Oncotype DX (ODX) assay, we acknowledge that a formal cost–benefit or health economic evaluation would strengthen the practical relevance of our findings. Multiple cost-effectiveness analyses have demonstrated that the Oncotype DX assay often yields additional quality-adjusted life years while reducing overall healthcare costs compared with using clinicopathologic criteria alone [30,31]. These gains are primarily driven by the avoidance of overtreatment with chemotherapy. While our study did not perform an economic evaluation, we propose that future work include a decision-analytic modeling framework to assess the nomogram’s impact on treatment costs, patient outcomes, and resource utilization within the Korean healthcare system.

Our study has some limitations. First, the inherent limitations of a retrospective study must be acknowledged. Prospective decision-impact studies—evaluating chemotherapy utilization, survival outcomes, and cost-effectiveness—will be essential to confirm the clinical utility of the nomogram in Korean patient. However, the retrospective design of this study allowed for the analysis of a large, homogeneous cohort of over 1200 patients who were treated under the same clinical protocol at a single institution. Second, the higher proportion of young patients with breast cancer in this study compared to that of the Tennessee model may have contributed to the observed differences [3]. The incidence of breast cancer in Korea is characterized by a higher prevalence of young patients, and this study reflects this epidemiological trend [32]. Third, a selection bias cannot be completely ruled out. At our institution, patients eligible for testing were selected according to the NCCN guidelines, and only those with the financial capacity to undergo the ODX test were selected. However, since we did not arbitrarily select patients but rather selected those who met the eligibility criteria within the study period, the likelihood of researcher-induced selection bias was minimal.

## 5. Conclusions

The Tennessee nomogram is useful when the actual ODX test is unavailable in the Korean population. However, in cases where a patient exhibits aggressive features, including a high histological grade, PR negativity, and high Ki-67, caution should be exercised when applying the Tennessee model. Since these characteristics are closely related to a higher probability of a high ODx risk score, the ODX test should be considered before using the nomogram. The use of a nomogram to identify suitable patients for the ODX test has the potential to reduce financial burden while maintaining treatment safety and precision.

## Figures and Tables

**Figure 1 cancers-17-03083-f001:**
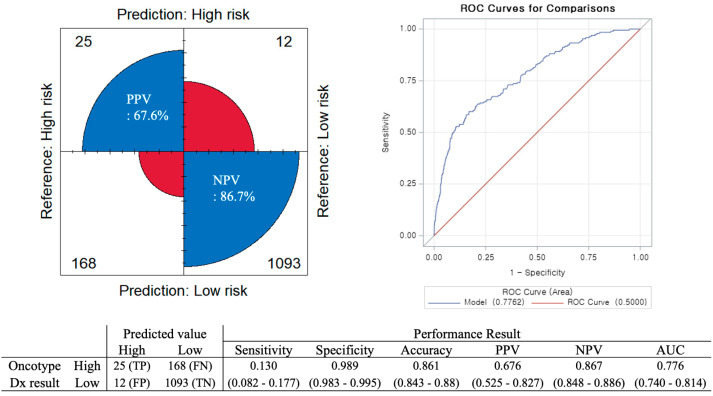
Tennessee nomogram performance in a Korean population. Abbreviations: ROC, Receiver Operating Characteristic; TP, True positive; FN, False negative; FP, False positive; TN, True negative; PPV, Positive predictive value; NPV, Negative predictive value; AUC, Area under the curve.

**Figure 2 cancers-17-03083-f002:**
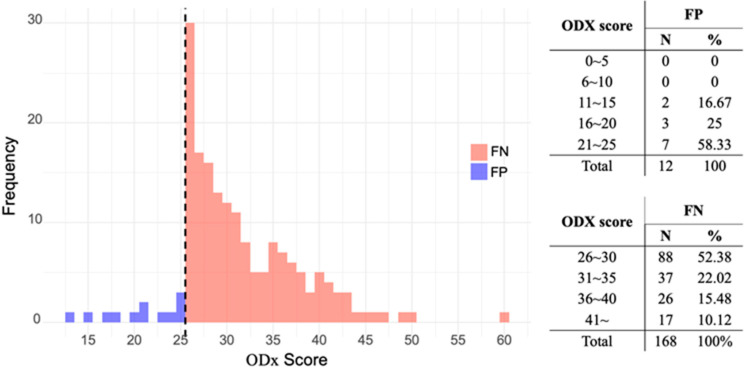
Distribution of Oncotype Dx (ODX) recurrence score in discordant groups with false negative and false positive results. Abbreviations: TP, True positive; FN, False negative; FP, False positive; TN, True negative; PPV, Positive predictive value; NPV, Negative predictive value; AUC, Area under the curve.

**Table 1 cancers-17-03083-t001:** Baseline clinicopathological characteristics of the study population.

	Total (n = 1298)	
	N	%
Age		
<50	683	52.6
≥50	615	47.4
T stage		
T1b	232	17.9
T1c	764	58.8
T2	302	23.3
N stage		
N0	1298	100.0
Histologic type		
IDC	1160	89.4
ILC	138	10.6
Histologic grade		
1	379	29.2
2	812	62.6
3	107	8.2
Progesterone receptor		
Negative	174	13.4
Positive	1124	86.6
HER2		
0	307	23.7
1+	600	46.2
2+, SISH no amplification	391	30.1
Ki-67		
<20%	907	70.1
≥20%	387	29.9
Missing	4	
Recurrence risk		
Low (≤25)	1105	85.1
High (≥25)	193	14.9

Abbreviations: IDC, invasive ductal carcinoma; ILC, invasive lobular carcinoma; HER2, human epidermal growth factor-2 receptor; SISH, silver in situ hybridization.

**Table 2 cancers-17-03083-t002:** Patient characteristics of false negative and true positive results.

	FN	(n = 168)	TP	(n = 25)	*p*
	N	*%*	N	*%*	
Age					0.504
<50	65	38.7	12	48.0	
≥50	103	61.3	13	52.0	
T stage					0.397
T1	131	78.0	17	68.0	
T2	37	22.0	8	32.0	
Histologic type					>0.999
IDC	156	92.9	24	96.0	
ILC	12	7.1	1	4.0	
Histologic grade					<0.001
1	17	10.1	0	0	
2	117	69.6	5	20.0	
3	34	20.3	20	80.0	
Progesterone receptor					<0.001
negative	46	27.4	24	96.0	
Positive	122	72.6	1	4.0	
HER2					0.339
0	40	23.8	4	16.0	
1+	66	39.3	8	32.0	
2+	62	36.9	13	52.0	
Ki-67					0.017
<20%	80	47.6	5	20.0	
≥20%	88	52.4	20	80.0	

Abbreviations: FN, False negative; TP, True positive; IDC, invasive ductal carcinoma; ILC, invasive lobular carcinoma; HER2, human epidermal growth factor-2 receptor.

**Table 3 cancers-17-03083-t003:** Patient characteristics of false negative and true negative results.

	FN	(n = 168)	TN	(n = 1093)	*p*
	N	*%*	N	*%*	
Age					<0.001
<50	65	38.7	601	55.0	
≥50	103	61.3	492	45.0	
T stage					0.154
T1	131	78.0	902	82.5	
T2	37	22.0	191	17.5	
Histologic type					0.102
IDC	156	92.9	969	88.6	
ILC	12	7.1	124	11.3	
Histologic grade					<0.001
1	17	10.1	362	33.1	
2	117	69.6	687	62.9	
3	34	20.3	44	4.0	
Progesterone receptor					<0.001
negative	46	27.4	93	8.5	
positive	122	72.6	1000	91.5	
HER2					0.053
0	40	23.8	260	23.8	
1+	66	39.3	523	47.6	
2+	62	36.9	310	28.4	
Ki-67					<0.001
<20%	80	47.6	818	75.1	
≥20%	88	52.4	271	24.9	

Abbreviations: FN, False negative; TN, True negative; IDC, invasive ductal carcinoma; ILC, invasive lobular carcinoma; HER2, human epidermal growth factor-2 receptor.

## Data Availability

The datasets generated and/or analyzed during the current study are not publicly available due to privacy restrictions but are available from the corresponding author on reasonable request.

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
