# Peer review of "Discrepancies Between the Tennessee Nomogram and Oncotype DX: Implications for the Korean Breast Cancer Population—The BRAIN Study"

_cancers, 2025, doi:10.3390/cancers17183083_

Round 1
Reviewer 1 Report
Comments and Suggestions for Authors
The authors conducted a retrospective study including ~ 1,300 patients with HR+/HER2- T1N0 breast cancers who underwent Oncotype DX test and compared the results with the prediction of the Tennessee Nomogram. The subject is quite interesting, and the cohort is large compared to other published external validation studies. In addition, the authors investigate factors associated with incorrectly classified patients. This manuscript provides indeed new scientific data, but has some
Major comments
- The nomogram predicts a % of high or low risk ODX recurrence score, i.e. a numeric value. For evaluating predictive performances of this test, the authors determined an cut-off for considering high-risk prediction but it is not clear how this was determined. In page 3 lines 110-113 it is stated that the cut-off probability was the same as the ODX recurrence score. Therefore, a positive test (Tennessee nomogram) was a probability of high risk ODX >25%? However, a 25% probability of high risk is not equivalent to a high RS in terms of clinical decision. This is an important methodological limitation to my opinion, and could explain why performances are better in the low-risk prediction compared to the high-risk. Alternatives could be to determine an optimal cut-off or to thoroughly justify this choice of 25% with a clinical practice point of view.
- The discussion has no references, and not enough comparison with the literature on the Tennessee Nomogram (for instance: 10.1007/s10549-023-07141-5, 10.4048/jbc.2018.21.2.222, 10.3389/fonc.2025.1586262) or other algorithms to predict ODX recurrence scores (4143/crt.2018.357), or to directly predict recurrences instead of RS (10.1038/s41523-024-00651-5).
- The authors should follow STROBE guidelines for retrospective cohort study methodology.
Minor comments
- I do not agree with the phrase in the introduction “This suggests that the ODX test is conducted in only approximately one-third of eligible patients with breast cancer in the United States and < 20% of eligible patients in European countries”. The references provided are outdated for the US and the one for Europe is not a population-level / real world study.
- Tables 1 and 2 could be improved for higher readability
- Page 4, lines 141-143: this should go into the discussion and not results
- Discussion: to my opinion, 86% accuracy compared to 85 and 89% is not really a discrepancy
Author Response
We sincerely thank the reviewers for their thoughtful and constructive comments. We have carefully revised the manuscript according to the suggestions. Below, we provide a point-by-point response to each comment, with corresponding revisions indicated in the revised manuscript.
Reviewer 1
The authors conducted a retrospective study including ~ 1,300 patients with HR+/HER2- T1N0 breast cancers who underwent Oncotype DX test and compared the results with the prediction of the Tennessee Nomogram. The subject is quite interesting, and the cohort is large compared to other published external validation studies. In addition, the authors investigate factors associated with incorrectly classified patients. This manuscript provides indeed new scientific data, but has some
Major comments
- The nomogram predicts a % of high or low risk ODX recurrence score, i.e. a numeric value. For evaluating predictive performances of this test, the authors determined an cut-off for considering high-risk prediction but it is not clear how this was determined. In page 3 lines 110-113 it is stated that the cut-off probability was the same as the ODX recurrence score. Therefore, a positive test (Tennessee nomogram) was a probability of high risk ODX >25%? However, a 25% probability of high risk is not equivalent to a high RS in terms of clinical decision. This is an important methodological limitation to my opinion, and could explain why performances are better in the low-risk prediction compared to the high-risk. Alternatives could be to determine an optimal cut-off or to thoroughly justify this choice of 25% with a clinical practice point of view.
Response: We sincerely thank for this important methodological comment. To clarify and avoid any potential confusion:
- The RS cutoff of 25 refers specifically to the actual numeric Oncotype DX recurrence score, as used in high-profile trials such as RxPONDER and TAILORx, and also adopted in the Tennessee nomogram (Orucevic et al. 2019).
- In contrast, the nomogram output is a probability (ranging from 0 to 1), and the Tennessee model uses 5 (i.e., 50%) as the probability threshold to classify patients into high- or low-risk groups.
To ensure clarity for readers, we have added this distinction explicitly to the Methods section of the manuscript.
“ The predicted probability for each patient was then dichotomized into high- versus low-risk groups using a 0.5 threshold, consistent with the original Tennessee Nomogram's classification approach.” (Method, page3, line 113-115)
- The discussion has no references, and not enough comparison with the literature on the Tennessee Nomogram (for instance: 1007/s10549-023-07141-5, 10.4048/jbc.2018.21.2.222, 10.3389/fonc.2025.1586262) or other algorithms to predict ODX recurrence scores (4143/crt.2018.357), or to directly predict recurrences instead of RS (10.1038/s41523-024-00651-5).
Response: We sincerely thank the reviewer for highlighting the need to enrich the Discussion by citing relevant literature on the Tennessee Nomogram and alternative prediction models. In response, we have carefully incorporated additional references. Following your recommendation, we have expanded the total number of references to over 30, thereby substantially improving both the breadth and depth of our literature context.
- The authors should follow STROBE guidelines for retrospective cohort study methodology.
Response: We thank the reviewer for this valuable suggestion. In response, we have revised the manuscript to improve the placement of content between the Methods and Discussion sections, and we also corrected the totals and enhanced the readability of the Results. We carefully re-checked all 22 items of the STROBE checklist to ensure that no essential elements were missing and made revisions where necessary. In addition, we have added relevant references in the Discussion as recommended.
Minor comments
- I do not agree with the phrase in the introduction “This suggests that the ODX test is conducted in only approximately one-third of eligible patients with breast cancer in the United States and < 20% of eligible patients in European countries”. The references provided are outdated for the US and the one for Europe is not a population-level / real world study.
Response: We appreciate the reviewer’s critical observation regarding the dated references used to support ODX test utilization rates in the U.S. and Europe. In response, we conducted a thorough literature review to locate more recent and representative data:
- A comprehensive U.S. analysis found that 48% of women with early-stage HR+/HER2– breast cancer received Oncotype DX testing, based on data from 2010–2017 in the National Cancer Database Nature.
- Another large-scale U.S. cohort study (2025) of 319,771 early-stage HR+/HER2– patients reported that 54% received ODX testing ASCO Publications.
- While we could not identify a comparable pan-European, population-level study on ODX utilization, evidence from Italy indicates that usage remains "quite scattered" in real-world practice PMC.
Accordingly, we have revised the Introduction to replace the previous estimates with these up-to-date figures, acknowledging that ODX testing is substantially more common in the U.S. than initially indicated, and that utilization in Europe remains variable and region-specific. (page2, lines 65-73)
“ Despite its proven clinical utility, the Oncotype DX (ODX) test remains costly (approximately $4,000), and although utilization rates in the United States have risen, they remain incomplete. Recent national data indicate that between 43% and 48% of eligible early-stage HR+/HER2– breast cancer patients underwent ODX testing in the U.S. between 2010 and 2017 [12-15]. In Europe, comprehensive population-level data are scarce; however, observational data from Italy suggest real-world use is significant yet variably implemented across regions [16]. Consequently, clinicians are motivated to develop alternative, more accessible tools to predict recurrence risk and inform treatment decisions in place of the standard ODX assay [17]. “
- Tables 1 and 2 could be improved for higher readability
Response: We have reformatted the section for improved clarity, with better alignment and a more structured presentation of the subgroups.
- Page 4, lines 141-143: this should go into the discussion and not results
Response: We have moved the statement from page 4, lines 141–143, to the Discussion section. (page 7, lines 183-189)
- Discussion: to my opinion, 86% accuracy compared to 85 and 89% is not really a discrepancy
Response: We thank for this comment. Our intention was not to emphasize the 3% difference in overall accuracy compared with the original Tennessee study, but rather to highlight the larger discrepancy of approximately 25% observed between the Oncotype DX results and the nomogram predictions in our cohort. We believe this point is better appreciated when considering the distribution of false negatives and false positives, which was the main focus of our analysis.
Conclusion
We thank the reviewers again for their valuable insights. We believe that the revisions—clarifying methodology, correcting errors, expanding comparisons with prior literature, and strengthening the discussion—have significantly improved the manuscript.
Reviewer 2 Report
Comments and Suggestions for Authors
This study addresses an important clinical issue: the accessibility and reliability of surrogate tools like the Tennessee nomogram in predicting Oncotype DX (ODX) recurrence risk scores for hormone receptor-positive, HER2-negative early-stage breast cancer patients in Korea. Given the high cost and limited availability of ODX testing globally, especially in resource-constrained settings, validation of alternative prediction models is highly relevant.
Points for Further Consideration:
- The reported low sensitivity (0.130) indicates the nomogram misses many high-risk patients, which could potentially lead to undertreatment if relied upon solely. While the high specificity (0.989) suggests reliable identification of low-risk cases, clinical decisions should carefully weigh these limitations to avoid adverse outcomes.
- Although validating the Tennessee nomogram in a Korean cohort is valuable, the study could discuss more explicitly whether genetic, environmental, or healthcare system differences might affect the nomogram’s calibration and performance compared to the original population it was developed in.
- Given the observed discrepancies, the study might explore or suggest modifications to the nomogram incorporating additional clinicopathologic markers (e.g., Ki-67, PR status) or genomic features tailored for the Korean population to improve predictive accuracy.
- The nomogram is a lower-cost alternative to ODX, an explicit consideration of cost-benefit analyses or the impact on patient outcomes and healthcare resource utilization would provide a more comprehensive view of its practical utility.
- Given the low sensitivity and the presence of false negatives in higher-risk but clinicopathologically less aggressive subgroups, elaborating on potential clinical consequences and strategies to mitigate under-treatment would be valuable.
- The higher proportion of younger patients than in the original Tennessee cohort suggests possible population-specific tumor biology; discussing how age and associated tumor features might influence nomogram accuracy would strengthen contextual understanding.
- The current retrospective design is informative but prospective validation—including impact on treatment decisions and outcomes—would be important to confirm the clinical utility of the nomogram in Korean patients.
Author Response
We sincerely thank the reviewers for their thoughtful and constructive comments. We have carefully revised the manuscript according to the suggestions. Below, we provide a point-by-point response to each comment, with corresponding revisions indicated in the revised manuscript.
Reviewer 2
This study addresses an important clinical issue: the accessibility and reliability of surrogate tools like the Tennessee nomogram in predicting Oncotype DX (ODX) recurrence risk scores for hormone receptor-positive, HER2-negative early-stage breast cancer patients in Korea. Given the high cost and limited availability of ODX testing globally, especially in resource-constrained settings, validation of alternative prediction models is highly relevant.
Points for Further Consideration:
- The reported low sensitivity (0.130) indicates the nomogram misses many high-risk patients, which could potentially lead to undertreatment if relied upon solely. While the high specificity (0.989) suggests reliable identification of low-risk cases, clinical decisions should carefully weigh these limitations to avoid adverse outcomes.
Response: We sincerely thank for this important comment. We fully acknowledge the limitation of low sensitivity and have carefully considered its clinical implications. In particular, we examined the characteristics of the false-negative group in greater detail and discussed factors (for example, higher histological grade, PR negative status, and higher Ki-67 levels) that should be taken into account when interpreting these results. - Although validating the Tennessee nomogram in a Korean cohort is valuable, the study could discuss more explicitly whether genetic, environmental, or healthcare system differences might affect the nomogram’s calibration and performance compared to the original population it was developed in.
Response: We sincerely thank the reviewer for this thoughtful comment. We agree that genetic, environmental, and healthcare system differences may influence the calibration and performance of the Tennessee nomogram when applied to Korean patients. Recent large-scale genomic analyses of East Asian breast cancers, such as the K-MASTER study, have indeed highlighted molecular and clinical disparities associated with age and ethnicity (e.g., differential prevalence of PIK3CA, ARID1A, and GATA3 mutations) compared with Western cohorts. However, it should be noted that these studies mainly included advanced or metastatic cases and investigated genes not directly involved in the Oncotype DX recurrence score algorithm. Thus, while these findings provide important context, they cannot be directly extrapolated to the early-stage, HR+/HER2– population eligible for Oncotype DX testing. Nevertheless, in response to the reviewer’s suggestion, we have cited this literature and expanded our Discussion to acknowledge potential biological and systemic factors that may affect external validity.
“East Asian breast cancers have been reported to exhibit distinct mutational patterns, including a higher prevalence of PIK3CA mutations and a lower frequency of GATA3 mutations [28].” (page7, lines 205-207) - Given the observed discrepancies, the study might explore or suggest modifications to the nomogram incorporating additional clinicopathologic markers (e.g., Ki-67, PR status) or genomic features tailored for the Korean population to improve predictive accuracy.
Response: We sincerely appreciate the reviewer’s insightful comment. As part of our future research, we plan to construct a revised version of the Tennessee nomogram that incorporates Ki-67 in order to enhance its predictive performance. Furthermore, we intend to investigate additional clinicopathologic and genomic variables that may be particularly relevant in Korean patients, with the goal of improving the accuracy and clinical applicability of the model. - The nomogram is a lower-cost alternative to ODX, an explicit consideration of cost-benefit analyses or the impact on patient outcomes and healthcare resource utilization would provide a more comprehensive view of its practical utility.
Response: We appreciate the reviewer’s suggestion to consider cost-benefit implications. While our current study did not include formal economic analysis, existing literature consistently demonstrates that Oncotype DX testing is often cost-effective or even cost-saving compared to clinical-pathologic risk tools alone—for example, achieving lower net costs and modest QALY gains in node-positive early breast cancer (Berdunov et al. 2023), and generating per-patient savings of €6,768–€13,125 in Dutch real-world models (de Jongh et al. 2022). Although our study focused on nomogram validation, these findings support the potential value of developing a lower-cost, locally tailored alternative. Going forward, we plan to incorporate health-economic modeling to assess the nomogram’s impact on treatment decisions, patient outcomes, and resource utilization in our healthcare setting.- Berdunov V, Laws E, et al. Cost-effectiveness analysis of the Oncotype DX Breast Recurrence Score test in node-positive early breast cancer: more effective (additional QALYs) at lower cost compared to clinical risk tools alone. Breast Cancer Res Treat. 2024. PubMed
- Berdunov V, et al. Economic evaluation: 21-gene assay more effective (0.17 QALYs) at lower cost (−£519) over lifetime compared to clinical risk alone. [Journal of Medical Economics]. 2022. PubMed
Given that the Tennessee nomogram is presented as a lower-cost alternative to the Oncotype DX (ODX) assay, we acknowledge that a formal cost-benefit or health economic evaluation would strengthen the practical relevance of our findings. Multiple cost-effectiveness analyses have demonstrated that the Oncotype DX assay often yields additional QALYs while reducing overall healthcare costs compared with using clinicopathologic criteria alone [1–2]. These gains are primarily driven by the avoidance of overtreatment with chemotherapy. While our study did not perform an economic evaluation, we propose that future work include a decision-analytic modeling framework to assess the nomogram’s impact on treatment costs, patient outcomes, and resource utilization within the Korean healthcare system.
- Given the low sensitivity and the presence of false negatives in higher risk but clinicopathologically less aggressive subgroups, elaborating on potential clinical consequences and strategies to mitigate under-treatment would be valuable.
Response: We sincerely thank the reviewer for this thoughtful comment. We fully agree that low sensitivity may result in missing patients with high genomic risk but clinicopathologically less aggressive features, raising concern for potential under-treatment. To address this limitation, we plan to incorporate Ki-67 into the nomogram as an additional variable. Although the present study was limited to analyzing the observed discrepancies and thus did not include this modification, our ultimate goal is to develop a revised model integrating Ki-67. This will allow us to assess changes in the size of the false-negative group and evaluate potential improvements in sensitivity. (Discussion, page7, line 215-218) - The higher proportion of younger patients than in the original Tennessee cohort suggests possible population-specific tumor biology; discussing how age and associated tumor features might influence nomogram accuracy would strengthen contextual understanding.
Response: We sincerely thank the reviewer for this valuable comment. In response, we have added a discussion on this issue (page 7, line 197-203), referring to relevant previous studies, to highlight how differences in age distribution and tumor biology may affect the predictive performance of the nomogram.
“This age-distribution shift can alter the pretest probability of high RS and the effect sizes of key predictors (e.g., grade, PR, Ki-67), potentially affecting both calibration and discrimination. Younger patients are more likely to present with higher proliferative indices, lower PR expression, and luminal B–like tumors, which could reduce the transportability of the model to our population [6,24-26]. Age-stratified calibration/discrimination analyses, reconsideration of thresholds, and—if needed—recalibration or model updating (including incorporation of Ki-67) would improve predictive accuracy and clinical utility [27]” - The current retrospective design is informative but prospective validation—including impact on treatment decisions and outcomes—would be important to confirm the clinical utility of the nomogram in Korean patients.
Response: We have emphasized that prospective validation studies are necessary to confirm the clinical utility of the Tennessee nomogram in the Korean setting (Discussion, page 8, lines 240–242).
We thank the reviewers again for their valuable insights. We believe that the revisions—clarifying methodology, correcting errors, expanding comparisons with prior literature, and strengthening the discussion—have significantly improved the manuscript.
Reviewer 3 Report
Comments and Suggestions for Authors
This manuscript forces on a new nomogram used for predicting breast cancer recurrence risk and the potential benefit of chemotherapy. I still have some questions listed below:
1 In Line 127, the numbers of patients are 1105 and 192 (the total number is not 1297).
2 In Table 1, ki-67:<20% , n=907,≥20%,n=387,the total patients are not 1298.
3 If the authors could provide survival data to compare the accuracy of Tennessee Nomogram and ODX, it will be more meaningful for clinical practice.
4 In Line 24-26, Tennessee Nomogram would underestimate patients with high tumor grade, lack of progesterone receptor expression, and high Ki-67 levels, which would account for quite many breast cancer patients. So, the practical value of this nomogram does not seem very high.
5 The introduction and discussion section of this article need revisions. In Line 69-74, the authors mention the cost of ODX and clinicians need alternative tools. But in conclusion part, for patients with some characteristics, ODX should be considered before Tennessee Nomogram. It seems this new nomogram can not be widely used.
Author Response
We sincerely thank the reviewers for their thoughtful and constructive comments. We have carefully revised the manuscript according to the suggestions. Below, we provide a point-by-point response to each comment, with corresponding revisions indicated in the revised manuscript.
Reviewer 3
This manuscript forces on a new nomogram used for predicting breast cancer recurrence risk and the potential benefit of chemotherapy. I still have some questions listed below:
- In Line 127, the numbers of patients are 1105 and 192 (the total number is not 1297).
Response: We apologize for these typographical errors. The numbers have been corrected throughout the manuscript. (192 -> 193)
- In Table 1, ki-67:<20% , n=907,≥20%,n=387,the total patients are not 1298.
Response: There were missing value in Ki67 (4). We added missing value count in Table 1.
- If the authors could provide survival data to compare the accuracy of Tennessee Nomogram and ODX, it will be more meaningful for clinical practice.
Response: Unfortunately, survival data were not available for this retrospective cohort. We acknowledge this limitation and have stated explicitly in the Discussion that prospective studies including survival outcomes are necessary.
- In Line 24-26, Tennessee Nomogram would underestimate patients with high tumor grade, lack of progesterone receptor expression, and high Ki-67 levels, which would account for quite many breast cancer patients. So, the practical value of this nomogram does not seem very high.
Response: Thank you for your insightful comment. We fully agree that the Tennessee Nomogram may underestimate risk in patients with high tumor grade, absence of progesterone receptor expression, or elevated Ki-67 levels. This limitation is indeed relevant in our cohort, where such biological features are not uncommon. As a next step, we plan to refine the model by incorporating Ki-67 and potentially other pathologic factors to improve its discriminatory ability. Our intention is not to dismiss the clinical usefulness of the nomogram, but rather to highlight its current limitations and the need for population-specific modifications to enhance its practical value in Korean patients.
- The introduction and discussion section of this article need revisions. In Line 69-74, the authors mention the cost of ODX and clinicians need alternative tools. But in conclusion part, for patients with some characteristics, ODX should be considered before Tennessee Nomogram. It seems this new nomogram can not be widely used.
Response: Thank you for pointing out this inconsistency. We also recognize that the manuscript may give somewhat conflicting messages across the introduction, discussion, and conclusion. In this study, we intended to highlight the economic advantages of using the nomogram by adding a paragraph in the Discussion (page 8, lines 229–238), thereby demonstrating that the Tennessee Nomogram could serve as a more affordable and accessible alternative in many clinical settings. At the same time, we wished to caution that overreliance on the nomogram may be unsafe for patients with high-risk biological features, and thus emphasized that in this subgroup the Tennessee Nomogram cannot fully replace ODX. In response to your comment, we have further clarified the description of economic aspects in both the introduction and discussion. As a result, we have revised the manuscript to clearly state that the nomogram should be regarded not as a universal substitute but as a complementary tool, while also specifying the clinical contexts in which it may be most appropriately applied. (Conclusion, page8, lines 255-262)
Conclusion
We thank the reviewers again for their valuable insights. We believe that the revisions—clarifying methodology, correcting errors, expanding comparisons with prior literature, and strengthening the discussion—have significantly improved the manuscript.